# Upper extremity joint tenderness as a practical indicator for assessing presenteeism in rheumatoid arthritis patients: A cross-sectional observational study

Ryota Naito[1,2], Masashi Taniguchi[1,2], Hideo Onizawa[1,3], Tomoya Nakajima[1,2], Kayo McCracken[1], Masato Mori[1], Ryosuke Hiwa[2], Takuji Nakamura[1], Akira Onishi[4], Shuichi Matsuda[5], Akio Morinobu[2], Shinji Hirose[1], Yutaka Shinkawa[1], Hisanori Umehara[1], Masao Tanaka[1,4]*

**1** Center for Rheumatic Diseases, Nagahama City Hospital, Shiga, Japan, **2** Department of Rheumatology and Clinical Immunology, Graduate School of Medicine, Kyoto University, Kyoto, Japan, **3** Immunology Medicine, Shiga General Hospital, Shiga, Japan, **4** Department of Advanced Medicine for Rheumatic Diseases, Graduate School of Medicine, Kyoto University, Kyoto, Japan, **5** Department of Orthopaedic Surgery, Graduate School of Medicine, Kyoto University, Kyoto, Japan

* masatana@kuhp.kyoto-u.ac.jp

## Abstract

### Objective

Rheumatoid arthritis (RA) causes chronic polyarthritis and joint dysfunction, reducing work productivity. This reduction is mainly due to presenteeism, characterized by impaired work performance despite being present at work. This study aims to investigate the impact of specific joint involvement, particularly in the upper extremities, on work disability in RA patients.

### Methods

Annual surveys assessing work disability were conducted among RA outpatients enrolled in the Nagahama Riumachi Cohort at Nagahama City Hospital, using the Work Productivity and Activity Impairment Questionnaire (WPAI). A multivariate regression analysis was performed to examine the cross-sectional and longitudinal associations between self-reported presenteeism and the tender joint count (TJC) in the extremities across two WPAI surveys.

### Results

The analysis included 201 patients, 52% of whom reported presenteeism. Cross-sectional analysis revealed a significant positive correlation between three or more TJCs of the upper extremity and presenteeism, with a regression coefficient (β) = 17.9 (95% confidence interval [CI]: 9.85–25.9). Among the joints evaluated, the sum

which permits unrestricted use, distribution, and reproduction in any medium, provided the original author and source are credited.

**Data availability statement:** All relevant data are within the article and its Supporting Information files.

**Funding:** This study received no specific funding. Data were obtained through routine clinical practice. The costs associated with manuscript preparation and submission were supported by the institutional resources of M. Tanaka's affiliated institution.

**Competing interests:** I have read the journal's policy and the authors of this article have the following competing interests: M. Tanaka and A. O. are affiliated with a department that receives financial support from two local governments in Japan (Nagahama City, Shiga and Toyooka City, Hyogo) and two pharmaceutical companies (Ayumi Pharmaceutical Corp. and Asahi Kasei Pharma Corp.). M. Tanaka has received research grants and/or speaker fees from AbbVie GK, Astellas Pharma Inc., Bristol-Myers Squibb Company, Chugai Pharmaceutical Co., Ltd., Daiichi Sankyo Co., Ltd., Eisai Co., Ltd., Nippon Zoki Pharmaceutical Co., Ltd., Pfizer Inc., Taisho Pharmaceutical Co., Ltd., Teijin Pharma, Ltd., and UCB Japan Co., Ltd. A. O. has received grants from Pfizer Inc., Bristol-Myers Squibb Company, and Advantest, as well as personal fees from Asahi Kasei Pharma Corp., Chugai Pharmaceutical Co., Ltd., Eli Lilly Japan K.K., Ono Pharmaceutical Co., Ltd., Mitsubishi Tanabe Pharma Corp., Takeda Pharmaceutical Co., Ltd., and Daiichi Sankyo Co., Ltd. S. M. has received speaking fees from Pfizer Inc., Chugai Pharmaceutical Co., Ltd., Asahi Kasei Pharma Corp., Eisai Co., Ltd., Mitsubishi Tanabe Pharma Corp., and Teijin Pharma Ltd. A. M. has received honoraria from AbbVie GK, Chugai Pharmaceutical Co., Ltd., Eli Lilly Japan K.K., Eisai Co., Ltd., Pfizer Inc., Bristol-Myers Squibb, Mitsubishi Tanabe Pharma Co., Astellas Pharma Inc., and Gilead Sciences Japan, and has received research grants from AbbVie GK, Asahi Kasei Pharma Corp., Chugai Pharmaceutical Co., Ltd., and Mitsubishi Tanabe Pharma Corp. R. H. has received research grants and/or speaker fees from AbbVie GK, Asahi Kasei Pharma Corp., Bristol-Myers Squibb Company, Daiichi Sankyo Co., Ltd., Eisai Co., Ltd., Eli Lilly Japan K.K., GSK plc, Kissei, Pfizer Inc., Mitsubishi Tanabe Pharma Co., and UCB Japan. There was no

of TJCs in the shoulder area (β = 9.55, CI: 5.39–13.7) and the fingers (β = 1.60, CI: 0.35–2.85) were significantly correlated with presenteeism. Additionally, change in presenteeism was significantly correlated with change in upper extremity TJCs (β = 1.41, CI: 0.05–2.77). No significant correlation was observed between lower extremity TJCs and presenteeism in these multivariate regression analyses.

## Conclusions

The upper extremity TJC is strongly associated with presenteeism in RA patients. Minimizing TJC in the upper extremities, particularly in the shoulders and fingers, could be important treatment goal to reduce work disability in RA patients.

---

## Introduction

Rheumatoid arthritis (RA) is a disease of unknown cause in which joint pain and destruction due to inflammation can lead to physical dysfunction. The prevalence of RA in developed countries is as high as 0.5–1%, and RA imposes an economic burden on individuals and society, even with the availability of biologic disease-modifying anti-rheumatic drugs (DMARDs) [1–3]. The cost of illness due to RA includes not only the direct costs of medical treatment, but also a considerable proportion of indirect costs, most of which are thought to be accounted for by work disability [3].

The loss of productivity due to work disability in RA patients is not due to absence from work (absenteeism) but to impairment while at work (presenteeism), and this is also the case in Japan [4,5]. Previous studies have shown that disease activity indicators, including tender joint count (TJC), as well as the measure of functional disability such as the Health Assessment Questionnaire Disability Index (HAQ-DI), are associated with work disability in RA patients [2,6]. In particular, strong evidence supports the association between functional disability and work disability in RA patients, with HAQ-DI being a well-established predictor of work impairments [2,6]. RA patients are particularly prone to work disability with physical labor and heavy work [1,2,6]. While the modernization of industry has alleviated the physical burden of labor, upper extremity disabilities, especially of the fingers, may still affect work disability due to the essential role of the upper extremities in operating various tools and machinery. However, to date, no studies have focused on the impact of involvement of specific joints on work disability in patients with RA. The impact of joint involvement on the HAQ-DI score differs by joint site, with finger symptoms having a lesser effect compared to other joints [7]. Relying solely on the HAQ-DI score may underestimate the impact of joint involvement on work disability for certain joints.

In this study, we hypothesized that upper extremity joint involvement has a significant impact on work disability in Japanese RA patients. To test this hypothesis, we analyzed the relationship between upper extremity joint involvement and work disability, adjusting for confounding factors, including HAQ-DI. We found that the upper extremity TJC could serve as a useful marker for predicting work disability in RA patients.

additional external funding received for this study. The authors have declared that no other competing interests exist.

## Methods

### Study design

This study was designed as a cross-sectional observational study, with additional longitudinal analyses included. Eligible patients were those participating in the Naga-hama Riumachi Cohort at Nagahama City Hospital who met the American College of Rheumatology (ACR) 1987 or ACR 2010 classification criteria for RA [8,9] and visited the hospital between 21 March 2017 and 20 February 2020. To assess the work disability of these participants, we collected their responses to the Work Productivity and Activity Impairment (WPAI) questionnaire [10,11]. Patients who were not working during the observation period or who had no medical records in the seven days prior to completing the WPAI questionnaire were excluded from the analysis.

### Ethical considerations

The study was approved by the Ethics Committee of Nagahama City Hospital (approval number: H28-38). Physicians conducted the study after obtaining verbal and written consent from patients.

### Data collection

Information such as RA disease duration and comorbidities were obtained from a questionnaire completed at the time of enrollment in the Nagahama Riumachi Cohort. A standardized visit data set including the HAQ-DI, TJC and swollen joint count (SJC) of the 70 joints (68 joints in the American College of Rheumatology core set plus two thumb carpometacarpal joints) [12], and blood test results were recorded at each patient visit. The WPAI questionnaire was used to assess each patient's work disability at the first visit of each year. The patient's medical record was consulted for the use of oral steroids and disease-modifying antirheumatic drugs (DMARDs). The change in each variable from the initial to the subsequent WPAI survey was quan-tified using the delta symbol (Δ). For instance, ΔCRP represented the difference in values for C-reactive protein (CRP) between the first and second surveys. The last access to the cohort database to collect the above data was made on November 27, 2020, and the data were anonymized at that time and recorded in a spreadsheet for various subsequent statistical analyses.

### TJC evaluation of upper and lower extremities

Of the 70 joints evaluated, 68 were classified as the upper and lower extremity joints, excluding the bilateral temporomandibular joints. The upper extremity joints con-sisted of 40 joints, including the bilateral sternoclavicular, acromioclavicular, shoulder, elbow, wrist, first carpometacarpal, metacarpophalangeal (MCP), thumb interphalan-geal (IP), proximal interphalangeal (PIP), and distal interphalangeal (DIP) joints. The lower extremity joints consisted of 28 joints, including the bilateral hip, knee, ankle, tarsal, metatarsophalangeal (MTP), and PIP joints. The TJC for the MCP, IP, PIP, and DIP joints of the hands, and for the MTP and PIP joints of the feet were summed to calculate the total TJC for each hand and foot. A higher TJC than SJC is associated

with greater physical dysfunction and work disability [13], and joint tenderness correlates with pain during motion [14]. Thus, we considered TJC more likely to affect work productivity than SJC and examined its relationship with work disability. In both descriptive and cross-sectional analyses, upper and lower extremity TJCs were categorized into three groups (0, 1–2, and >2) with cutoffs set at 0 and 2, as the upper extremity TJC was 0 in more than half of the patients, and the median TJC was 2 in patients with an upper extremity TJC greater than 0. The same cutoffs were set for lower extremity TJC. A model was also created with two categories (0 and > 0) for both upper and lower extremity TJC. Changes in the upper extremities (Δupper extremity TJC) between the two WPAI survey time points were categorized as "increased" when greater than 0, "no change" when equal to 0, and "decreased" when less than 0. Additionally, we investigated the association between tender joint regions and work disability. Assuming joint function is a key contributor, we analyzed joints as functional units rather than as individual anatomical joints by summing the TJC values as continuous variables. Specifically, we defined the total shoulder TJC as the sum of the TJCs for both sternoclavicular joints, acromioclavicular joints, and shoulder joints; the total finger TJC as the sum of the TJCs for the MCP, IP, PIP, DIP, and first carpometacarpal joints in both hands; and the total foot TJC as the sum of the TJCs for the ankle, tarsal, MTP, and PIP joints in both feet.

### WPAI questionnaire

WPAI questionnaire was used to assess work disability due to RA [10,11]. The WPAI consists of the following six questions (Q1–Q6) about work disability in the past 7 days: Q1 = currently employed; Q2 = hours missed due to RA; Q3 = hours missed due to other reasons; Q4 = hours actually worked; Q5 = degree to which RA affected productivity while working (using a 0–10-cm visual analog scale, VAS); and Q6 = degree to which RA affected productivity in regular unpaid activities (using a 0–10-cm VAS). The four main outcomes were quantified as percentages by multiplying the following scores by 100: 1) work time missed due to RA (absenteeism) = Q2/(Q2 + Q4), 2) percent impairment while working due to RA (presenteeism) = Q5/10,3) percent overall work impairment due to RA (overall work impairment) = Q2/(Q2 + Q4)+(1-Q2)/(Q2 + Q4)×(Q5/10), 4) percent activity impairment due to RA (activity impairment) = Q6/10.

### Statistical analysis

Baseline data were summarized using descriptive statistics, with means and standard deviations or medians and interquartile ranges for continuous variables, and proportions for categorical variables. The association between work disability and the upper extremity TJC, as well as TJC in each area of the extremities, was analyzed using univariate and multivariate linear regression models. Covariates were selected from factors reported or estimated to be associated with the upper extremity TJCs and work disability including age, gender, disease duration, HAQ-DI, erythrocyte sedimentation rate (ESR), oral steroids [1,2,6], bDMARDs [5,15,16], and lower extremity TJCs. Since targeted synthetic DMARDs (tsDMARDs) suppress disease activity similarly to bDMARDs [17], they were combined into a single variable, "bDMARDs or tsDMARDs". In the linear regression analyses using TJC in each area of extremities, the hip joint was excluded due to a 1% prevalence. Additionally, we examined the impact of changes in the total upper extremity TJC and TJC in each area of the extremities on change in presenteeism, using WPAI data at two time points and employing univariate and multivariate linear regression models. The same covariates were used as in the cross-sectional analysis except that oral steroid reduction or discontinuation was defined as "reduction of oral steroids", and bDMARDs or tsDMARDs initiation as "introduction of bDMARDs or tsDMARDs". Due to less than 2% of patients showing change in TJC, the hip joint was excluded from the linear regression analyses. In the multivariate analyses, the variance inflation factors for all independent variables were below 5. P values less than 0.05 were considered significant. Significance levels were assigned as follows: ns, not significant; * p < 0.05; ** p < 0.01; *** p < 0.001. Statistical analyses were performed using JMP Pro 15 (SAS Institute), except for the Jonckheere-Terpstra test performed using R version 4.3.2 (R Development Core Team, Vienna, Austria).

## Results

### Baseline characteristics and joint tenderness distribution

There were 483 RA patients in the cohort who visited a hospital during the observation period, with WPAI data obtained from 471 of these patients. After excluding 258 non-working patients and 12 patients lacking medical records within seven days of completing the WPAI data, 201 RA patients remained eligible (Fig 1). Table 1 presents the baseline characteristics of the 201 patients analyzed at their first response to WPAI questionnaire. The mean age was 55.0 years, with 19.4% of the patients older than 65 years, and 65.7% were female. The median disease duration was 4.0 years, the median HAQ-DI was 0.1, and the median Disease Activity Score based on 28 joints and ESR (DAS28-ESR) was 2.8. Conventional synthetic DMARDs were used in 83.6%, bDMARDs in 20.3%, and tsDMARDs in 1.5% (Table 1). Table 2 shows the frequency of TJC across various anatomical joints. Tenderness was observed in 23.4% of the patients on either or both sides of the finger joints (IP, MCP, PIP, and DIP joints), 15.9% for the wrist joints, 10.4% for the shoulder joints, 4.0% for the elbow joints, 10.9% for the knee joints, 8.0% for the ankle joints, and 8.0% for the toe joints (MTP and PIP joints). Tenderness in the sternoclavicular, acromioclavicular, first carpometacarpal, hip, and tarsal joints was observed in 1.5% or less of patients (Table 2).

### Prevalence of presenteeism in the RA patients

At the time of the first WPAI questionnaire, absence from work (absenteeism) due to RA symptoms was observed in only 7.5% of patients, while decreased productivity at work (presenteeism) was observed in more than half of the patients (52.2%). Overall work impairment due to RA symptoms was present in 53.7% of patients. Impairment in activities of daily living other than work (activity impairment) was present in 60.7% of patients (S1 Fig).

### Analysis of the relationship between upper extremity TJC and presenteeism

We next examined the association between upper extremity TJC and work disability. As upper extremity TJC increased, WPAI outcomes scores tended to increase (Fig 2).

To further explore the relationship, we examined various demographic and clinical indicators as shown in Table 3. Using a linear regression model, we analyzed the association of presenteeism with demographic characteristics and clinical indicators of RA. For two patients, ESR test results could not be referenced in the medical record, and they were excluded from the

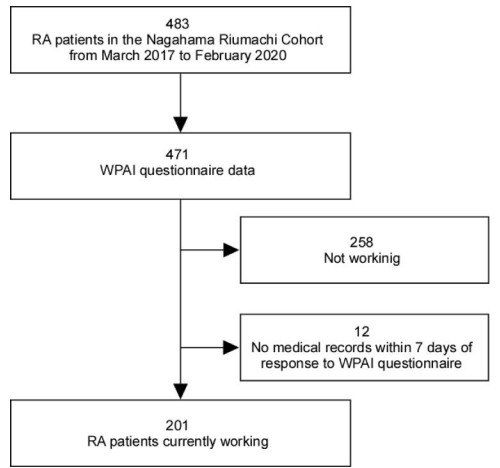

**Fig 1. Diagram of participant flow.**

**Table 1. Patient characteristic.**

| Characteristics | All | Upper extremity TJC | | |
|---|---|---|---|---|
| | | 0 | 1-2 | >2 |
| Patient, no (%) | 201 (100) | 128 (63.7) | 39 (19.4) | 34 (16.9) |
| Age (years), mean±SD | 55.0±11.8 | 55.6±12.0 | 54.4±12.4 | 53.7±10.3 |
| ≥65 years, no (%) | 39 (19.4) | 27 (21.1) | 8 (20.5) | 4 (11.8) |
| Gender | | | | |
| Female, no (%) | 132 (65.7) | 90 (70.3) | 19 (48.7) | 23 (67.6) |
| Male, no (%) | 69 (34.3) | 38 (30.0) | 20 (51.3) | 11 (32.4) |
| Anti-CCP antibody positive, no (%) | 150 (80.2) | 93 (77.5) | 29 (80.6) | 28 (90.3) |
| Disease duration (years), median (Q1-Q3) | 4.0 (1.0-10.0) | 6.0 (2.0-10.0) | 3.0 (1.0-9.0) | 1.0 (0.0-5.0) |
| HAQ-DI, median (Q1-Q3) | 0.1 (0-0.5) | 0 (0-0.4) | 0.3 (0-0.6) | 0.5 (0.1-0.6) |
| DAS28-ESR, median (Q1-Q3) | 2.8 (1.7-3.5) | 2.1 (1.5-2.9) | 3.3 (2.8-4.0) | 4.8 (3.8-5.9) |
| CDAI, median (Q1-Q3) | 5.6 (2.1-10.7) | 2.8 (1.6-6.2) | 9.0 (6.1-12.1) | 17.4 (12.7-25.8) |
| SDAI, median (Q1-Q3) | 6.0 (2.3-11.0) | 2.9 (1.7-6.3) | 9.6 (6.8-13.0) | 19.0 (12.7-26.8 |
| ESR (mm/h), median (Q1-Q3) | 16.0 (8.0-29.0) | 13.0 (6.0-24.3) | 16.0 (8.0-28.0) | 31.5 (15-56) |
| CRP (mg/dl), median (Q1-Q3) | 0.1 (0.0-0.4) | 0.1 (0.0-0.2) | 0.2 (0.1-0.9) | 0.4 (0.1-1.4) |
| TJC of 70 joints, median (Q1-Q3) | 0 (0-2) | 0 (0−0) | 2 (1-3) | 8 (4.8-12) |
| Lower extremity TJC 0, no (%) | 156 (77.6) | 115 (89.8) | 28 (71.8) | 13 (38.2) |
| Lower extremity TJC 1–2, no (%) | 30 (14.9) | 10 (7.8) | 8 (20.5) | 12 (35.3) |
| Lower extremity TJC>2, no (%) | 15 (7.5) | 3 (2.3) | 3 (7.7) | 9 (26.5) |
| SJC of 70 joints, median (Q1-Q3) | 0 (0-1.5) | 0 (0−0) | 1 (0-2) | 3 (0-7.5) |
| Upper extremity SJC 0, no (%) | 140 (69.7) | 112 (87.5) | 16 (41.0) | 12 (35.3) |
| Upper extremity SJC 1–2, no (%) | 32 (15.9) | 10 (7.8) | 17 (43.6) | 5 (14.7) |
| Upper extremity SJC>2, no (%) | 29 (14.4) | 6 (4.7) | 6 (15.4) | 17 (50.0) |
| Lower extremity SJC 0, no (%) | 179 (89.1) | 121 (94.5) | 33 (84.6) | 25 (73.5) |
| Lower extremity SJC 1–2, no (%) | 13 (6.5) | 5 (3.9) | 4 (10.3) | 4 (11.8) |
| Lower extremity SJC>2, no (%) | 9 (4.5) | 2 (1.6) | 2 (5.1) | 5 (14.7) |
| Comorbidities | | | | |
| Interstitial pneumonia, no (%) | 18 (9.0) | 11 (8.6) | 5 (12.8) | 2 (5.9) |
| Overlap with other CTDs, no (%) | 13 (6.5) | 8 (6.3) | 3 (7.7) | 2 (5.9) |
| Current medication | | | | |
| Oral steroid, no (%) | 55 (27.4) | 29 (22.7) | 11 (28.2) | 15 (44.1) |
| csDMARDs, no (%) | 168 (83.6) | 107 (83.6) | 36 (92.3) | 25 (73.5) |
| bDMARDs, no (%) | 41 (20.3) | 34 (26.6) | 3 (7.7) | 4 (11.8) |
| tsDMARDs, no (%) | 3 (1.5) | 2 (1.6) | 1 (2.6) | 0 (0.0) |

SD: standard deviation; Q1: 1st Quartile; Q3: 3rd Quartile; HAQ-DI: Health Assessment Questionnaire Disability Index; DAS28-ESR: Disease Activity Score based on 28 joints and erythrocyte sedimentation rate; CDAI: Clinical Disease Activity Index; SDAI: Simplified Disease Activity Index; ESR: erythrocyte sedimentation rate; CRP: C-reactive protein; TJC: tender joint count; SJC: swollen joint count; CTDs: connective tissue diseases including mixed connective tissue disease, polymyositis and dermatomyositis, Sjögren's syndrome, systemic lupus erythematosus and systemic sclerosis; csDMARDs: conventional synthetic disease-modifying anti-rheumatic drugs; bDMARDs: biological disease-modifying anti-rheumatic drugs; tsDMARDs: targeted synthetic disease-modifying anti-rheumatic drugs.

analysis using ESR and DAS28-ESR. In the univariate regression analysis, the HAQ-DI and composite disease activity measures for RA such as DAS28-ESR were significantly associated with presenteeism. Furthermore, TJC and SJC of 70 joints were significantly associated with presenteeism. Next, we analyzed the specific impact of affected joints in the upper and lower extremities on presenteeism. "Upper extremity TJC 1-2" and "upper extremity TJC >2" were significantly associated

**Table 2. Prevalence of tender joints in the patients.**

| Tender joint sites | All | Upper extremity TJC | | |
|---|---|---|---|---|
| | | 0 | 1-2 | >2 |
| Patient, no (%) | 201 (100) | 128 (63.7) | 39 (19.4) | 34 (16.9) |
| Temporomandibular joint | | | | |
| Both or either side, no (%) | 2 (1.0) | 1 (0.8) | 0 (0.0) | 1 (2.9) |
| Right, no (%) | 2 (1.0) | 1 (0.8) | 0 (0.0) | 1 (2.9) |
| Left, no (%) | 1 (0.5) | 1 (0.8) | 0 (0.0) | 0 (0.0) |
| Upper extremity joints | | | | |
| Sternoclavicular joint | | | | |
| Both or either side, no (%) | 2 (1.0) | 0 (0.0) | 1 (2.6) | 1 (2.9) |
| Right, no (%) | 0 (0.0) | 0 (0.0) | 0 (0.0) | 0 (0.0) |
| Left, no (%) | 2 (1.0) | 0 (0.0) | 1 (2.6) | 1 (2.9) |
| Acromioclavicular joint | | | | |
| Both or either side, no (%) | 3 (1.5) | 0 (0.0) | 2 (5.1) | 1 (2.9) |
| Right, no (%) | 1 (0.5) | 0 (0.0) | 0 (0.0) | 1 (2.9) |
| Left, no (%) | 3 (1.5) | 0 (0.0) | 2 (5.1) | 1 (2.9) |
| Shoulder joint | | | | |
| Both or either side, no (%) | 21 (10.4) | 0 (0.0) | 7 (17.9) | 14 (41.2) |
| Right, no (%) | 17 (8.5) | 0 (0.0) | 6 (15.4) | 11 (32.4) |
| Left, no (%) | 19 (9.5) | 0 (0.0) | 6 (15.4) | 13 (38.2) |
| Elbow joint | | | | |
| Both or either side, no (%) | 8 (4.0) | 0 (0.0) | 3 (7.7) | 5 (14.7) |
| Right, no (%) | 5 (2.5) | 0 (0.0) | 1 (2.6) | 4 (11.8) |
| Left, no (%) | 7 (3.5) | 0 (0.0) | 3 (7.7) | 4 (11.8) |
| Wrist joint | | | | |
| Both or either side, no (%) | 32 (15.9) | 0 (0.0) | 12 (30.8) | 20 (58.8) |
| Right, no (%) | 25 (12.4) | 0 (0.0) | 9 (23.1) | 16 (47.1) |
| Left, no (%) | 24 (11.9) | 0 (0.0) | 8 (20.5) | 16 (47.1) |
| First carpometacarpal joint | | | | |
| Both or either side, no (%) | 3 (1.5) | 0 (0.0) | 1 (2.6) | 2 (5.9) |
| Right, no (%) | 0 (0.0) | 0 (0.0) | 0 (0.0) | 0 (0.0) |
| Left, no (%) | 3 (1.5) | 0 (0.0) | 1 (2.6) | 2 (5.9) |
| IP, MCP, PIP (hand) and DIP joints | | | | |
| Both or either side, no (%) | 47 (23.4) | 0 (0.0) | 16 (12.5) | 31 (91.2) |
| Right, no (%) | 38 (19.0) | 0 (0.0) | 11 (28.2) | 27 (79.4) |
| Left, no (%) | 32 (15.9) | 0 (0.0) | 6 (15.4) | 26 (76.5) |
| TJC for IP, MCP, PIP (hand) and DIP joints | | | | |
| Both sides, median (Q1-Q3) | 0 (0–0) | 0 (0–0) | 0 (0-1) | 4 (2-8) |
| Right, median (Q1-Q3) | 0 (0–0) | 0 (0–0) | 0 (0-1) | 2 (1-4) |
| Left, median (Q1-Q3) | 0 (0–0) | 0 (0–0) | 0 (0–0) | 2 (0.8-4) |
| Lower extremity joints | | | | |
| Hip joint | | | | |
| Both or either side, no (%) | 2 (1.0) | 0 (0.0) | 1 (2.6) | 1 (2.9) |
| Right, no (%) | 2 (1.0) | 0 (0.0) | 1 (2.6) | 1 (2.9) |
| Left, no (%) | 1 (0.5) | 0 (0.0) | 0 (0.0) | 1 (2.9) |
| Knee joint | | | | |
| Both or either side, no (%) | 22 (10.9) | 7 (5.5) | 6 (15.4) | 9 (26.5) |

*(Continued)*

**Table 2.** (Continued)

| Tender joint sites | All | Upper extremity TJC | | |
|---|---|---|---|---|
| | | 0 | 1-2 | >2 |
| Right, no (%) | 17 (8.5) | 5 (3.9) | 5 (12.8) | 7 (20.6) |
| Left, no (%) | 18 (9.0) | 5 (3.9) | 4 (10.3) | 9 (26.5) |
| Ankle joint | | | | |
| Both or either side, no (%) | 16 (8.0) | 4 (3.1) | 5 (12.8) | 7 (20.6) |
| Right, no (%) | 13 (6.5) | 3 (2.3) | 5 (12.8) | 5 (14.7) |
| Left, no (%) | 12 (6.0) | 3 (2.3) | 3 (7.7) | 6 (17.6) |
| Tarsal joint | | | | |
| Both or either side, no (%) | 3 (1.5) | 2 (1.6) | 0 (0.0) | 1 (2.9) |
| Right, no (%) | 1 (0.5) | 0 (0.0) | 0 (0.0) | 1 (2.9) |
| Left, no (%) | 3 (1.5) | 2 (1.6) | 0 (0.0) | 1 (2.9) |
| MTP and PIP (foot) joints | | | | |
| Both or either side, no (%) | 16 (8.0) | 2 (1.6) | 2 (5.1) | 12 (35.3) |
| Right, no (%) | 15 (7.5) | 2 (1.6) | 2 (5.1) | 11 (32.4) |
| Left, no (%) | 9 (4.5) | 0 (0.0) | 2 (5.1) | 7 (20.6) |
| TJC for MTP and PIP (foot) joints | | | | |
| Both sides, median (Q1-Q3) | 0 (0–0) | 0 (0–0) | 0 (0–0) | 0 (0-1.3) |
| Right, median (Q1-Q3) | 0 (0–0) | 0 (0–0) | 0 (0–0) | 0 (0-1) |
| Left, median (Q1-Q3) | 0 (0–0) | 0 (0–0) | 0 (0–0) | 0 (0–0) |

IP: interphalangeal; MCP: metacarpophalangeal; PIP: proximal interphalangeal; DIP: distal interphalangeal; MTP: metatarsophalangeal. See Table 1 for other abbreviation definitions.

with presenteeism compared to "upper extremity TJC 0". Notably, "upper extremity TJC > 2" showed higher regression coefficients than "upper extremity TJC 1-2". Similarly, "lower extremity TJC 1-2" and "lower extremity TJC >2" were significantly associated with presenteeism in contrast to "lower extremity TJC 0". The results were similar to upper and lower extremity SJC, but upper and lower extremity SJC were significantly associated with presenteeism when the affected joint was greater than 2, respectively. Multivariate regression analysis revealed that the regression coefficient (β) and 95% confidence interval (CI) for upper extremity TJC on presenteeism were as follow: β=4.45 (CI: −2.1–11.0) for "TJC 1-2" and β=17.9 (CI: 9.85–25.9) for "TJC>2," showing the latter was statistically significant. This result suggests that the level of work disability rises in tandem with the increasing number of affected joints. On the other hand, lower extremity TJC was not significantly associated with presenteeism (Table 3). When the upper and lower extremity TJCs were analyzed as binary variables, considering the presence or absence of tender joints, comparable results were observed, specifically showing that upper extremity TJC had an association with presenteeism, whereas lower extremity TJC did not (S1 Table). These results suggest that RA-related joint damage, particularly in the upper extremity, affects work disability.

## Analysis of the relationship between shoulder and finger TJC and presenteeism

We next examined the joints significantly contributing to work disability. Table 4 shows the association between presenteeism and TJC of each area of the extremities, using linear regression models. In the univariate regression analysis, the TJC of the total shoulder, wrist, total finger, knee, and total foot joints were significantly associated with presenteeism. In the multivariate regression analysis, the total shoulder TJC was significantly associated with presenteeism (β=9.55, CI: 5.39–13.7), as was the total finger TJC (β=1.60, CI: 0.35–2.85) (Table 4). These results suggest that involvement of the shoulder and finger joints in RA patients contributes to presenteeism.

 

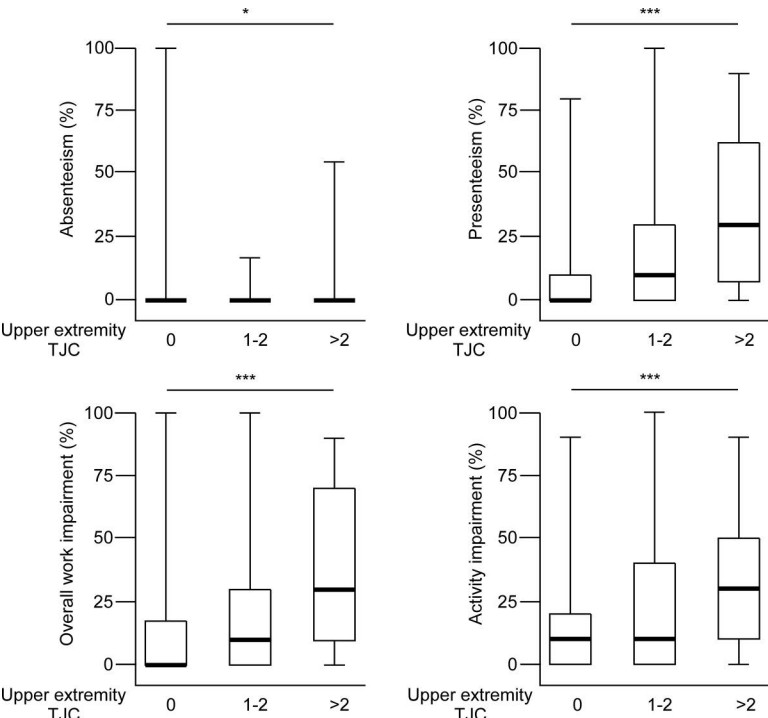

**Fig 2. Relationship between upper extremity TJC and WPAI outcomes.** Patients are categorized into three groups by upper extremity TJC: 0 (n = 128), 1-2 (n = 39), and > 2 (n = 34). The median (bold line), 25th percentile, 75th percentile (box), and range (whiskers) are shown. The vertical axis represents the WPAI outcomes expressed as percentages. The significance of the differences in WPAI outcomes among the three groups was tested using the Jonckheere-Terpstra test.

## Analysis of the relationship between temporal changes in upper extremity TJC and presenteeism

We examined the association between change in upper extremity TJC over time and change in presenteeism. Within the observation period, WPAI data were collected twice for the same patient in 158 cases. Of these, 137 were continuing to work and 21 were retired. The retired group was significantly older than the working group (64.0 ± 9.8 vs. 53.5 ± 11.6 years, mean ± standard deviation) (Fig 3A). The average interval between the two WPAI surveys for the working group was 248 ± 121 days (mean ± standard deviation). In the working group, presenteeism decreased between the first and second WPAI surveys (Fig 3B). Of the 137 working patients, we analyzed 134, excluding three due to missing medical records within 7 days of the WPAI survey. Presenteeism due to RA tended to improve as upper extremity TJC improved (Fig 3C).

To further quantify these observations, we used a linear regression model to examine the relationship between changes in clinical indicators of RA and presenteeism, with results shown in Tables 5 and 6. In the univariate regression analysis, we found that changes in total upper extremity TJC, as well as total shoulder joint TJC, wrist joint TJC, and HAQ-DI were significantly associated with change in presenteeism (Tables 5 and 6). Multivariate regression analysis identified changes in upper extremity TJC (β = 1.41, CI: 0.05–2.77) and HAQ-DI (β = 45.3, CI: 32.4–58.2) as significant independent variables that correlated with change in presenteeism (Table 5). Another multivariate regression analysis using changes in TJC for each area of the extremities as separate independent variables yielded β = 5.19 (CI: −0.77–11.1, p = 0.087) for total shoulder TJC, and β = 1.23 (CI: −0.40–2.85, p = 0.138) for total finger TJC, suggesting a trend toward correlation with the change in presenteeism (Table 6). These results suggest that change in upper extremity TJC is associated with change in work disability.

**Table 3. The association of demographic and clinical indicators with the percentage of presenteeism.**

| Independent variables | Univariate | | Multivariate (n = 199) | |
|---|---|---|---|---|
| | β value estimate (95% CI) | *P* | β value estimate (95% CI) | *P* |
| Age (years) | −0.18 (−0.44, 0.08) | 0.168 | −0.23 (−0.45, −0.01) | 0.038 |
| Male vs female | 1.74 (−4.70, 8.18) | 0.596 | 3.55 (−2.02, 9.12) | 0.210 |
| Anti-CCP antibody positive | 1.88 (−5.94, 9.71) | 0.636 | | |
| Disease duration (years) | 0.07 (−0.33, 0.47) | 0.742 | 0.32 (−0.02, 0.66) | 0.064 |
| HAQ-DI | 27.2 (21.8, 32.6) | <0.001 | 23.6 (17.9, 29.4) | <0.001 |
| DAS28-ESR | 7.26 (5.30, 9.23) | <0.001 | | |
| CDAI | 1.33 (1.01, 1.64) | <0.001 | | |
| SDAI | 1.28 (0.98, 1.57) | <0.001 | | |
| ESR (mm/h) | 0.27 (0.12, 0.41) | <0.001 | 0.03 (−0.10, 0.16) | 0.658 |
| CRP (mg/dl) | 7.16 (3.87, 10.4) | <0.001 | | |
| TJC of 70 joints | 1.55 (1.00, 2.10) | <0.001 | | |
| Upper extremity TJC | | | | |
|   0 | Reference | | | |
|   1-2 | 9.11 (1.92, 16.3) | 0.013 | 4.45 (−2.10, 11.0) | 0.182 |
|   >2 | 25.4 (17.8, 33.0) | <0.001 | 17.9 (9.85, 25.9) | <0.001 |
| Lower extremity TJC | | | | |
|   0 | Reference | | | |
|   1-2 | 13.0 (4.59, 21.4) | 0.003 | −0.12 (−7.40, 7.15) | 0.973 |
|   >2 | 14.0 (2.60, 25.3) | 0.011 | −0.72 (−10.8, 9.40) | 0.888 |
| SJC of 70 joints | 2.27 (1.44, 3.11) | <0.001 | | |
| Upper extremity SJC | | | | |
|   0 | Reference | | | |
|   1-2 | 2.71 (−5.35, 10.8) | 0.508 | | |
|   >2 | 20.5 (12.1, 28.9) | <0.001 | | |
| Lower extremity SJC | | | | |
|   0 | Reference | | | |
|   1-2 | 11.8 (−0.09, 23.7) | 0.052 | | |
|   >2 | 29.8 (15.6, 43.9) | <0.001 | | |
| Comorbidities | | | | |
|   Interstitial pneumonia | −3.18 (−13.9, 7.53) | 0.559 | | |
|   Overlap with other CTDs | −2.77 (−15.2, 9.66) | 0.661 | | |
| Current medication | | | | |
|   Oral steroid | 5.46 (−1.36, 12.3) | 0.116 | −0.21 (−5.86, 5.44) | 0.941 |
|   bDMARDs or tsDMARDs | 0.60 (−6.81, 8.00) | 0.874 | −2.64 (−8.96, 3.68) | 0.411 |

Multivariate model: $R^2$ = 0.44. CI: confidence interval. See Table 1 for other abbreviation definitions.

## Discussion

More than half of RA patients reported presenteeism, while absenteeism was infrequently observed. This result is consistent with a previous report from Japan [5], suggesting that presenteeism, rather than absenteeism, predominantly contributes to work disability in Japan [3,4].

Work disability in RA patients correlated with both composite and non-composite measures of disease activity and HAQ-DI, aligning with previous reports [2,6,18], suggesting that inflammation and dysfunction of joints cause work disability. Although

**Table 4. The association of joint tenderness distribution with the presenteeism percentage.**

| Independent variables | Univariate | | Multivariate (n = 199) | |
|---|---|---|---|---|
| | β value estimate (95% CI) | *P* | β value estimate (95% CI) | *P* |
| Upper extremity, both sides | | | | |
| Total shoulder TJC | 16.1 (11.7, 20.5) | <0.001 | 9.55 (5.39, 13.7) | <0.001 |
| Elbow joint TJC | 6.84 (−2.99, 16.7) | 0.172 | −1.22 (−8.81, 6.37) | 0.751 |
| Wrist joint TJC | 12.0 (7.13, 16.9) | <0.001 | 1.36 (−3.26, 5.98) | 0.562 |
| Total finger TJC | 2.49 (1.49, 3.49) | <0.001 | 1.60 (0.35, 2.85) | 0.013 |
| Lower extremity, both sides | | | | |
| Knee joint TJC | 11.0 (5.30, 16.6) | <0.001 | 0.21 (−4.90, 5.31) | 0.937 |
| Total foot TJC | 1.86 (0.28, 3.44) | 0.021 | −1.05 (−2.83, 0.73) | 0.247 |

Multivariate model: $R^2$ = 0.48. CI: confidence interval. The values in the multivariate analysis are adjusted for the confounding effects of the following variables: age (years), gender, disease duration (years), HAQ-DI, ESR (mm/h), oral steroid, and bDMARDs or tsDMARDs. Total shoulder TJC: the sum of the TJC values for the sternoclavicular, acromioclavicular, and shoulder joints. Total finger TJC: the sum of the TJC values for the first carpometacarpal, metacarpophalangeal, interphalangeal, proximal interphalangeal (hand), and distal interphalangeal joints. Total foot TJC: the sum of the TJC values for the ankle, tarsal, metatarsophalangeal, and proximal interphalangeal (foot) joints. See Table 1 for other abbreviation definitions.

previous reports have identified TJC as a factor associated with work disability in RA patients [2,6], the specific distribution of affected joints contributing to presenteeism has not been extensively studied. Our data showed that total upper extremity TJC, total shoulder joint TJC, and total finger joint TJC are significant independent predictors for presenteeism in RA patients, even after adjusting for confounders. Joint tenderness is known to be associated with pain during movement [14], and pain has been shown to be an important predictor of work disability in RA patients [6,19,20]. These findings suggest that TJC reflects movement-related pain, which may contribute to work disability. Less manual dexterity is associated with work disability in RA patients [21]. Work disability is prevalent in other conditions, such as systemic sclerosis (SSc), characterized by skin hardening, contractures, and peripheral vascular complications. SSc patients with digital ulcers have higher hand dysfunction and an increased risk of work disability [22–24]. Hand dermatitis has also been reported to cause functional impairment of hand and work disability [25]. These findings are consistent with our analysis, providing objective evidence that finger disorders contribute to work disability. In our data, total finger joint TJC emerged as a significant predictor of work disability even after adjusting for the HAQ-DI, an established predictors of work disability in RA [2,6,18]. Symptoms related to the finger joints have a lesser impact on HAQ-DI [7], suggesting that this measure may not fully capture critical impairments in hand function related to work. Notably, finger joint TJC may reveal aspects of work disability inadequately captured by HAQ-DI. On the other hand, the specific contribution of shoulder involvement in RA to work disability has not yet been adequately investigated. Shoulder lesions, regardless of the underlying disease, have been estimated to reduce working life by 1.8 to 8.1 years compared to the general population in Finland [26], suggesting that shoulder lesions significantly impact work disability. In RA, shoulder joint involvement exerts the greatest influence on the HAQ-DI among all the 68 joints included in the ACR core set, contributing 28.3% to the total score [27], highlighting its significant impact on physical function. Our findings are the first to demonstrate that the association between shoulder lesions and work disability in RA remains significant even after adjusting for the HAQ-DI. This suggests that, similar to the observations regarding finger joint involvement, HAQ-DI alone does not provide a comprehensive assessment of work disability associated with shoulder joint lesions in RA. Therefore, work disability in patients with RA should be evaluated not only through the HAQ-DI but also considering upper extremity TJC, particularly the TJC of the shoulder and finger joints, as important indicators.

Furthermore, change in upper extremity TJC was a significant independent variable for change in presenteeism, even after adjusting for confounders. While the correlation between changes in total shoulder TJC and presenteeism, as well as changes in total finger joint TJC and presenteeism, did not reach statistical significance, there may be a trend toward a positive correlation. These findings suggest that upper extremity TJC could serve as a valuable indicator of change in work disability.

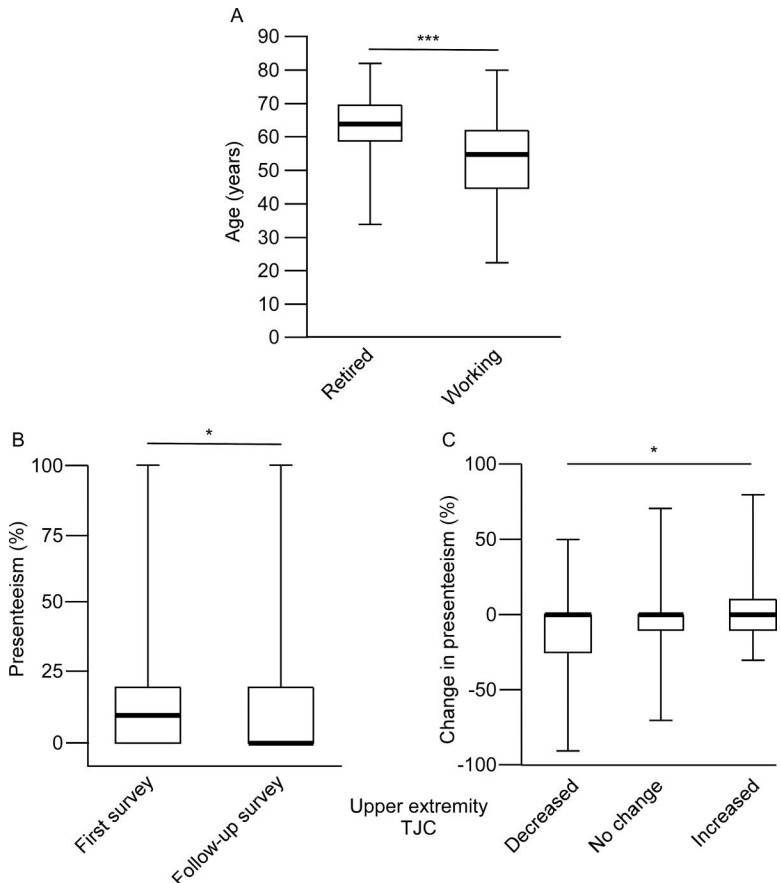

**Fig 3. Characteristics of patients who continued working during the two WPAI surveys.** A: Age comparison between retired (n = 21) and working (n = 137) patients who completed two WPAI surveys (n = 158). B: Presenteeism (%) at the first and second WPAI surveys for the 137 patients who continued working. C: Relationship between changes in upper extremity TJC and presenteeism, with patients grouped by upper extremity TJC changes: decreased (n = 37), unchanged (n = 73), and increased (n = 27). Median (bold line), 25th/75th percentiles (box), and range (whiskers) are shown. The vertical axis shows age at the first WPAI survey in A, presenteeism (%) in B, and the change in presenteeism (%) in **C.** Differences were determined by the Student's t-test in A, the Wilcoxon signed-rank test in B, and the Jonckheere-Terpstra test in **C.**

This study has several limitations. First, the data were not adjusted for socioeconomic factors (such as educational level, job type and income), psychological condition [2,6], or the patient's subjective pain level [19,20], all of which have been reported to affect work disability and were inadequately considered. Further studies incorporating these factors are needed. Second, the analysis included a substantial number of patients who had been undergoing treatment for RA for an extended period, and was restricted to those currently employed. Long-term RA patients may not experience significant improvement in disease activity with treatment, and some patients may have transitioned to careers that are more accommodating to their disability or have completely retired from workforce [28]. Therefore, our results may underestimate treatment-related improvements in work disability and the overall impact of RA on work disability. Third, retirees identified between the two WPAI surveys were excluded from the analysis of changes in presenteeism over time. The average age of these retired patients was 64 years, likely associated with the typical Japanese retirement age. However, work disability related to age or disease activity may have contributed to their retirement. Fourth, as a limitation of the study design, the cross-sectional nature of this study limits the ability to establish causality. Additionally, this study was conducted at a single center in Japan. Further investigation is necessary to generalize these findings to a broader RA patient population.

**Table 5. The relationship between the changes in the clinical indicators and the change in the percentage of presenteeism.**

| Independent variables | Univariate | | Multivariate (n = 134) | |
|---|---|---|---|---|
| | β value estimate (95% CI) | P | β value estimate (95% CI) | P |
| Age (years) | −0.12 (−0.47, 0.24) | 0.510 | −0.02 (−0.36, 0.32) | 0.927 |
| Male vs female | −4.04 (−13.0, 4.87) | 0.371 | −1.34 (−9.73, 7.05) | 0.753 |
| Disease duration (years) | −0.24 (−0.82, 0.35) | 0.419 | −0.48 (−0.99, 0.03) | 0.064 |
| ΔHAQ-DI | 45.1 (33.1, 57.1) | <0.001 | 45.3 (32.4, 58.2) | <0.001 |
| ΔDAS28-ESR | 6.29 (3.46, 9.13) | <0.001 | | |
| ΔCDAI | 1.02 (0.60, 1.44) | <0.001 | | |
| ΔSDAI | 0.97 (0.66, 1.37) | <0.001 | | |
| ΔESR (mm/h) | 0.30 (0.04, 0.56) | 0.027 | 0.05 (−0.19, 0.30) | 0.677 |
| ΔCRP (mg/dl) | 3.36 (−1.28, 7.99) | 0.154 | | |
| ΔTJC of 70 joints | 0.89 (−0.02, 1.80) | 0.055 | | |
| Δupper extremity TJC | 1.36 (0.08, 2.63) | 0.037 | 1.41 (0.05, 2.77) | 0.043 |
| Δlower extremity TJC | 1.15 (−1.05, 3.35) | 0.304 | −1.36 (−3.59, 0.87) | 0.231 |
| ΔSJC of 70 joints | 1.24 (0.20, 2.28) | 0.020 | | |
| Δupper extremity SJC | 1.73 (0.50, 2.96) | 0.006 | | |
| Δlower extremity SJC | 0.41 (−2.41, 3.22) | 0.775 | | |
| Reduction of oral steroid dose | −2.67 (−13.1, 7.80) | 0.615 | 2.99 (−6.43, 12.4) | 0.531 |
| Introduction of bDMARDs or tsDMARDs | −7.85 (−20.6, 4.86) | 0.224 | −0.12 (−11.4, 11.2) | 0.984 |

Multivariate model: $R^2$ = 0.34. Δ: Difference between the values at the first and second WPAI survey.

See Table 1 for other abbreviation definitions.

**Table 6. The relationship between changes in TJCs and changes in percentage of presenteeism across assessed joints.**

| Independent variables | Univariate | | Multivariate (n = 134) | |
|---|---|---|---|---|
| | β value estimate (95% CI) | P | β value estimate (95% CI) | P |
| Upper extremity, both sides | | | | |
| Δtotal shoulder jonit TJC | 11.5 (5.67, 17.4) | <0.001 | 5.19 (−0.77, 11.1) | 0.087 |
| Δelbow joint TJC | −1.5 (−12.2, 9.19) | 0.782 | −5.76 (−15.2, 3.63) | 0.227 |
| Δwrist joint TJC | 12.1 (5.53, 18.7) | <0.001 | 4.87 (−1.71, 11.5) | 0.146 |
| Δtotal finger TJC | 0.58 (−0.88, 2.03) | 0.433 | 1.23 (−0.40, 2.85) | 0.138 |
| Lower extremity, both sides | | | | |
| Δknee joint TJC | 3.91 (−3.90, 11.7) | 0.324 | −3.71 (−10.8, 3.42) | 0.305 |
| Δtotal foot TJC | 0.98 (−1.40, 3.36) | 0.416 | −0.99 (−3.47, 1.49) | 0.431 |

Multivariate model: $R^2$ = 0.37. The values in the multivariate analysis are adjusted for the confounding effects of the following variables: age (years), gender, disease duration (years), ΔHAQ-DI, ΔESR (mm/h), reduction of oral steroid dose, and introduction of bDMARDs or tsDMARDs. See Tables 1, 4 and 5 for other abbreviation definitions.

## Conclusions

This study is the first to evaluate the relationship between the distribution of joint involvement and work disability in RA patients. Our findings highlight the upper extremity TJC, particularly in the shoulders and fingers, as a significant predictor of RA-related work disability. Presenteeism due to RA tended to improve as upper extremity TJC improved, suggesting that minimizing TJC in the upper extremities could be an important treatment goal to reduce work disability in RA patients.

## Supporting information

**S1 Fig. Distribution of WPAI outcomes at the first survey.** The vertical axis shows the percentages of patient, with each WPAI outcome displayed separately and shaded in grayscale.
(TIF)

**S1 Table. The association of upper and lower extremity TJC as binary variables with the percentage of presenteeism.**
(DOCX)

**S1 File. The minimal anonymized dataset necessary to replicate the study findings.**
(XLSX)

## Acknowledgments

We would like to thank all the patients who participated in the present study. We would also like to acknowledge Y. Yoshinaga, H. Hosokawa, M. Fukuda and W. Yamamoto for their secretarial assistance.

## Author contributions

**Conceptualization:** Ryota Naito, Akira Onishi, Masao Tanaka.

**Data curation:** Ryota Naito, Masashi Taniguchi, Hideo Onizawa, Tomoya Nakajima, Kayo McCracken, Masato Mori, Ryosuke Hiwa, Takuji Nakamura, Shinji Hirose, Yutaka Shinkawa, Hisanori Umehara, Masao Tanaka.

**Formal analysis:** Ryota Naito.

**Funding acquisition:** Masao Tanaka.

**Investigation:** Ryota Naito, Masao Tanaka.

**Methodology:** Ryota Naito, Akira Onishi, Masao Tanaka.

**Project administration:** Masao Tanaka.

**Resources:** Masao Tanaka.

**Software:** Ryota Naito.

**Supervision:** Akira Onishi, Shuichi Matsuda, Akio Morinobu, Hisanori Umehara, Masao Tanaka.

**Validation:** Ryota Naito.

**Visualization:** Ryota Naito.

**Writing – original draft:** Ryota Naito.

**Writing – review & editing:** Masao Tanaka.

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
