## [Decision Letter · Decision Letter 0]

25 Mar 2025

Dear Dr. Tanaka,

Thank you for submitting your manuscript to PLOS ONE. After careful consideration, we feel that it has merit but does not fully meet PLOS ONE’s publication criteria as it currently stands. Therefore, we invite you to submit a revised version of the manuscript that addresses the points raised during the review process.

**Kindly look into the editor's and reviewers comments and revise the manuscript accordingly.**

We look forward to receiving your revised manuscript.

Kind regards,

Sham Santhanam

Academic Editor

PLOS ONE

**Journal Requirements:**

1. When submitting your revision, we need you to address these additional requirements. Please ensure that your manuscript meets PLOS ONE's style requirements, including those for file naming. The PLOS ONE style templates can be found at https://journals.plos.org/plosone/s/file?id=wjVg/PLOSOne_formatting_sample_main_body.pdf and https://journals.plos.org/plosone/s/file?id=ba62/PLOSOne_formatting_sample_title_authors_affiliations.pdf 2. Thank you for stating in your Funding Statement: M.Tanaka and A.O. belong to the department financially supported by two local governments in Japan (Nagahama City, Shiga and Toyooka City, Hyogo) and two pharmaceutical companies (Ayumi Pharmaceutical Corp. and Asahi Kasei Pharma Corp.).M.Tanaka received research grants and/or speaker fees from AbbVie GK, Astellas Pharma Inc., Bristol-Myers Squibb Company, Chugai Pharmaceutical Co., Ltd., Daiichi Sankyo Co., Ltd., Eisai Co., Ltd., Nippon Zoki Pharmaceutical Co., Ltd., Pfizer Inc., Taisho Pharmaceutical Co., Ltd., Teijin Pharma, Ltd. and UCB Japan Co., Ltd. A.O. received grants from Pfizer Inc., Bristol-Myers Squibb Company, and Advantest, as well as personal fees from Asahi Kasei Pharma Corp., Chugai Pharmaceutical Co., Ltd., Eli Lilly Japan K.K., Ono Pharmaceutical Co., Ltd., Mitsubishi Tanabe Pharma Corp., Takeda Pharmaceutical Co., Ltd. and Daiichi Sankyo Co., Ltd. S.M. received speaking fees from Pfizer Inc., Chugai Pharmaceutical Co. Ltd., Asahi Kasei Pharma Corp., Eisai Co. Ltd., Mitsubishi Tanabe Pharma Corp., and Teijin Pharma Ltd.; A.M. received honoraria from AbbVie G.K., Chugai Pharmaceutical Co. Ltd., Eli Lilly Japan K.K., Eisai Co. Ltd., Pfizer Inc., Bristol–Myers Squibb, Mitsubishi Tanabe Pharma Co., Astellas Pharma Inc. and Gilead Sciences Japan, and received research grants from AbbVie G.K., Asahi Kasei Pharma Corp., Chugai Pharmaceutical Co. Ltd., and Mitsubishi Tanabe Pharma Corp.R.H. received research grants and/or speaker fees from AbbVie GK, Asahi Kasei Pharma Corp., Bristol Myers Squibb Company, Daiichi Sankyo Co., Ltd., Eisai Co., Ltd., Eli Lilly Japan K.K., GSK plc, Kissei, Pfizer Inc., Mitsubishi Tanabe Pharma Co., and UCB Japan.  Please provide an amended statement that declares *all* the funding or sources of support (whether external or internal to your organization) received during this study, as detailed online in our guide for authors at http://journals.plos.org/plosone/s/submit-now.  Please also include the statement “There was no additional external funding received for this study.” in your updated Funding Statement. Please include your amended Funding Statement within your cover letter. We will change the online submission form on your behalf.

**Additional Editor Comments:**

Congrats on the effort.

1. Mention the type of study - cross sectional/observational - in the title and in methods

Please refer to standard reporting guidelines and the checklist (STROBE - if it's an observational study)

2. The methods section can be trimmed further - section on TJC evaluation and statistical analysis

3. In results, please mention the individual subheadings without inferences (not to mention if it has a positive or negative correlation)

4. Not to repeat the content in tables/figures extensively. Use figures only if really needed (figure 2)

5. Also look into the reviewers comments.

Reviewers' comments:

Reviewer's Responses to Questions

**Comments to the Author**

1. Is the manuscript technically sound, and do the data support the conclusions?

Reviewer #1: Yes

Reviewer #2: Yes

2. Has the statistical analysis been performed appropriately and rigorously?

Reviewer #1: Yes

Reviewer #2: Yes

3. Have the authors made all data underlying the findings in their manuscript fully available?

Reviewer #1: Yes

Reviewer #2: Yes

4. Is the manuscript presented in an intelligible fashion and written in standard English?

Reviewer #1: Yes

Reviewer #2: Yes

**Reviewer #1:**  I congratulate the authors for a well conducted study. The study carries a message that has clinical implications. I would like to suggest some minor changes

Comment 1

The abstract does not reflect the study completely. Please include the following lines in abstract

1. Please mention in methods that presenteeism was ‘self-reported’

2. Please mention in the results that a positive correlation was seen between upper extremity TJC and presenteeism but not with lower extremity TJC

3. Please change the last line in conclusion of abstract to ‘Minimizing TJC in the

upper extremities, particularly in the shoulders and fingers, could be important

treatment goal to reduce work disability in RA patients.’

Comment 2

Please summarise the study in conclusion in the manuscript in a better way

Comment 3

Please mention the average interval between collecting WPAI data for 158 patients that were included in the temporal change comparison.

**Reviewer #2:**  This cross-sectional study provides valuable insights into work disability among patients with rheumatoid arthritis. The manuscript is well-written, and the title, abstract, and introduction are appropriate and adequately presented.

However, the cross-sectional design limits the ability to establish causality. To strengthen the analysis, the authors may consider incorporating a multivariable model that adjusts for socioeconomic status, as this could be an important confounder in assessing work disability.

Overall, this is an impressive manuscript that contributes to the existing literature on the subject.

**Do you want your identity to be public for this peer review?** For information about this choice, including consent withdrawal, please see our Privacy Policy

Reviewer #1: No

Reviewer #2: **Yes: ** Vijaya Prasanna Parimi

---

## [Author Response · Author response to Decision Letter 1]

28 Apr 2025

Response to Editor’s comments

We sincerely appreciate editor’s encouraging words and the opportunity to improve our manuscript. As outlined below, we have addressed the editor's comments with utmost sincerity and care.

Comment 1

Mention the type of study - cross sectional/observational - in the title and in methods.

Please refer to standard reporting guidelines and the checklist (STROBE - if it's an observational study).

Response: We have revised the title (Page 1, Line 1) and the “Methods” section (Page 6, Lines 3-4) to specify the study type. Additionally, since our study includes certain longitudinal analyses, we have also mentioned this in the “Methods” section (Page 6, Lines 3–4). Furthermore, we have referred to the STROBE checklist and confirmed that our manuscript adheres to these reporting guidelines.

Comment 2

The methods section can be trimmed further - section on TJC evaluation and statistical analysis.

Response: We have trimmed the sections “TJC evaluation of upper and lower extremities” (Page 7-9) and “Statistical analysis” (Page 10, after the line, and Page 11) by reducing the number of words by changing the wording (465 words => 375 words, and 372 words => 305 words, respectively), while ensuring accuracy.

Comment 3

In results, please mention the individual subheadings without inferences (not to mention if it has a positive or negative correlation).

Response: We have revised the “Results” section to ensure that the subheadings do not include inferences (Pages 13-16).

Comment 4

Not to repeat the content in tables/figures extensively. Use figures only if really needed (figure 2).

Response: We have revised the “Results” section to retain the key points related to the tables and figures by not mentioning duplicate items (Pages 12-17, deleted items are in red, double lined text). Additionally, following the editor’s suggestion, we have determined that Figure 2 is not essential for the main text and have therefore moved it to the Supporting Information (S1 Fig). We have also updated the figure numbering throughout the manuscript accordingly (in the red letters after the “Results” section).

Comment 4

Also look into the reviewer comments.

Response: We have addressed Reviewer #1’s and Reviewer #2’s comments as well. Responses to their comments can be found in the next section.

Response to reviewers’ comments

Reviewer #1:

We sincerely thank Reviewer #1 for the positive feedback and for recognizing the clinical implications of our study. We appreciate the constructive suggestions and have addressed them accordingly.

Comment 1

The abstract does not reflect the study completely. Please include the following lines in abstract

1. Please mention in methods that presenteeism was ‘self-reported’

2. Please mention in the results that a positive correlation was seen between upper extremity TJC and presenteeism but not with lower extremity TJC

3. Please change the last line in conclusion of abstract to ‘Minimizing TJC in the

upper extremities, particularly in the shoulders and fingers, could be important

treatment goal to reduce work disability in RA patients.’

Response: As suggested by Reviewer #1, we have revised the abstract as follows:

1. We have clarified that presenteeism was ‘self-reported’ in the “Methods” section of the “Abstract” (Page 2, Line 11).

2. We have revised the “Results” section of the “Abstract” to indicate that a positive correlation was seen between upper extremity TJC and presenteeism, but not with lower extremity TJC (Page 3, Lines 2–3).

3. We have modified the last sentence of the “Conclusion” of the “Abstract” to: ‘Minimizing TJC in the upper extremities, particularly in the shoulders and fingers, could be an important treatment goal to reduce work disability in RA patients.’ (Page 3, Lines 5 and 6).

Comment 2

Please summarise the study in conclusion in the manuscript in a better way.

Response: As suggested by Reviewer #1, we have revised the conclusion to provide a clearer and more concise summary of our findings. Specifically, we have emphasized the significance of upper extremity TJC, particularly in the shoulders and fingers, as a predictor of work disability in RA patients.

Comment 3

Please mention the average interval between collecting WPAI data for 158 patients that were included in the temporal change comparison.

Response: Reviewer #1 mentioned 158 patients; however, the temporal change analysis included 137 patients who continued working, after excluding 21 retirees. Therefore, we have reported the average interval for the 137 patients as 248 ± 121 days in the manuscript and have also made slight modifications to the surrounding text for clarity and coherence (Page 16, Lines 10 and 11).

Reviewer #2:

We sincerely thank Reviewer #2 for the positive feedback and for recognizing the value of our study. We greatly appreciate Reviewer #2’s kind words regarding the title, abstract, and introduction.

Specific point

However, the cross-sectional design limits the ability to establish causality. To strengthen the analysis, the authors may consider incorporating a multivariable model that adjusts for socioeconomic status, as this could be an important confounder in assessing work disability. Overall, this is an impressive manuscript that contributes to the existing literature on the subject.

Response: We sincerely appreciate Reviewer #2’s valuable comments, particularly regarding the potential confounding effect of socioeconomic status (SES) on work disability. As we mentioned in the "Discussion" section (the first limitation comment on Page 20, Lines 13-17), we acknowledge this limitation and the need for cautious interpretation of our results. Unfortunately, our cohort data does not include indicators of SES which have been reported as relevant factors in the association with work disability. Additionally, due to the lack of ethical approval to collect SES data from the facility, we were unable to incorporate this information into our analysis. However, although adjustments for SES as a confounder are not included, we believe that our data still offer meaningful insights into the relationship between upper extremity TJC and work disability in RA patients, contributing to a broader understanding of this topic.

Furthermore, as suggested by Reviewer #2 regarding the limitations of the cross-sectional study design, we have revised the manuscript to clearly acknowledge this issue. Specifically, we have noted that the cross-sectional nature of our study limits the ability to establish causality and requires careful interpretation (Page 22, Lines 9-11).

We sincerely appreciate your insightful suggestions, and we hope that our responses adequately address your concerns.

---

## [Decision Letter · Decision Letter 1]

12 May 2025

Upper extremity joint tenderness as a practical indicator for assessing presenteeism in rheumatoid arthritis patients: A cross-sectional observational study

PONE-D-24-60059R1

Dear Dr. Tanaka,

We’re pleased to inform you that your manuscript has been judged scientifically suitable for publication and will be formally accepted for publication once it meets all outstanding technical requirements.

Kind regards,

Martin Feuchtenberger

Academic Editor

PLOS ONE

Additional Editor Comments (optional):

Congratulations to the authors and many thanks also to the reviewers of the manuscript!

Kind regards,

Martin Feuchtenberger

Reviewers' comments:

Reviewer's Responses to Questions

**Comments to the Author**

Reviewer #1: All comments have been addressed

Reviewer #2: All comments have been addressed

2. Is the manuscript technically sound, and do the data support the conclusions?

Reviewer #1: Yes

Reviewer #2: Yes

3. Has the statistical analysis been performed appropriately and rigorously?

Reviewer #1: Yes

Reviewer #2: Yes

4. Have the authors made all data underlying the findings in their manuscript fully available?

Reviewer #1: Yes

Reviewer #2: Yes

5. Is the manuscript presented in an intelligible fashion and written in standard English?

Reviewer #1: Yes

Reviewer #2: Yes

Reviewer #1: Thank you for making the suggested corrections. Please find some minor comments

1. Please mention the units of the time interval between two WAPI surveys, whether it is days or weeks.

2. Please include this text in conclusion “Presenteeism due to RA tended to improve as upper extremity TJC improved”

Reviewer #2: This is a revised manuscript addressing the reviewer comments. The queries were answered. No specific extra comments to the author or editor.

**Do you want your identity to be public for this peer review?** For information about this choice, including consent withdrawal, please see our Privacy Policy

Reviewer #1: No

Reviewer #2: **Yes: ** Vijaya Prasanna Parimi

---

## [Editor Report · Acceptance letter]

PONE-D-24-60059R1

PLOS ONE

Dear Dr. Tanaka,

I'm pleased to inform you that your manuscript has been deemed suitable for publication in PLOS ONE. Congratulations! Your manuscript is now being handed over to our production team.

Kind regards,

on behalf of

Dr. Martin Feuchtenberger

Academic Editor

PLOS ONE